# MMP2 Modulates Inflammatory Response during Axonal Regeneration in the Murine Visual System

**DOI:** 10.3390/cells10071672

**Published:** 2021-07-02

**Authors:** Lien Andries, Luca Masin, Manuel Salinas-Navarro, Samantha Zaunz, Marie Claes, Steven Bergmans, Véronique Brouwers, Evy Lefevere, Catherine Verfaillie, Kiavash Movahedi, Lies De Groef, Lieve Moons

**Affiliations:** 1Neural Circuit Development and Regeneration Research Group, Department of Biology, KU Leuven, 3000 Leuven, Belgium; lien.andries@kuleuven.be (L.A.); luca.masin@kuleuven.be (L.M.); manual.salinas@um.es (M.S.N.); marie.claes@kuleuven.be (M.C.); steven.bergmans@kuleuven.be (S.B.); veronique.brouwers@kuleuven.be (V.B.); evy.lefevere@kuleuven.be (E.L.); lies.degroef@kuleuven.be (L.D.G.); 2Leuven Brain Institute, 3000 Leuven, Belgium; 3Department of Development and Regeneration, Stem Cell Institute, KU Leuven, 3000 Leuven, Belgium; samantha.zaunz@kuleuven.be (S.Z.); catherine.verfaillie@kuleuven.be (C.V.); 4Myeloid Cell Immunology Lab, VIB, Center for Inflammation Research, 1090 Brussels, Belgium; kiavash.movahedi@vub.be; 5Laboratory of Cellular and Molecular Immunology, Department of Biology, Vrije Universiteit Brussel, 1050 Brussels, Belgium

**Keywords:** matrix metalloproteinases, optic nerve injury, myeloid cells, axonal regeneration, inflammatory stimulation, expression profiling

## Abstract

Neuroinflammation has been put forward as a mechanism triggering axonal regrowth in the mammalian central nervous system (CNS), yet little is known about the underlying cellular and molecular players connecting these two processes. In this study, we provide evidence that MMP2 is an essential factor linking inflammation to axonal regeneration by using an in vivo mouse model of inflammation-induced axonal regeneration in the optic nerve. We show that infiltrating myeloid cells abundantly express MMP2 and that MMP2 deficiency results in reduced long-distance axonal regeneration. However, this phenotype can be rescued by restoring MMP2 expression in myeloid cells via a heterologous bone marrow transplantation. Furthermore, while MMP2 deficiency does not affect the number of infiltrating myeloid cells, it does determine the coordinated expression of pro- and anti-inflammatory molecules. Altogether, in addition to its role in axonal regeneration via resolution of the glial scar, here, we reveal a new mechanism via which MMP2 facilitates axonal regeneration, namely orchestrating the expression of pro- and anti-inflammatory molecules by infiltrating innate immune cells.

## 1. Introduction

Neurodegenerative diseases and central nervous system (CNS) trauma are largely irreversible, in part because adult mammals lack robust axonal regeneration potential [1,2,3]. Strikingly, neuroinflammation seems to induce axonal regrowth in adult mammalian CNS. It is increasingly clear that blood-borne and resident inflammatory cells as well as reactivated macroglia affect axonal regeneration [4,5,6,7]. One of the molecular families that has been implicated in axonal regeneration is that of matrix metalloproteinases (MMPs). More specifically, several studies within the visual system suggest a role for MMP2 in injury-induced axonal regrowth [1,8,9,10,11,12].

Despite its role in axonal regeneration being addressed in a multitude of studies, controversy remains about the expression profile of MMP2 in the visual system. This seems to be dependent on the examined species, the developmental stage, and/or whether ex vivo or in vivo approaches were followed. In primates, MMP2 was shown to be expressed in the retinal ganglion cell (RGC) stomata and their axons but not in macroglia [13,14,15], while in developing and adult zebrafish, MMP2 was detected in both the RGC axons and in retinal Müller glia [16]. Of note, MMP2 expression/activity was found to dramatically increase after optic nerve injury in adult fish [17,18,19]. Additionally, in mice, MMP2 is upregulated in outgrowing RGC axons and their growth cones during development [1,10,11,16]. In retinal explants of the adult mouse retina, MMP2 was found to be expressed in both RGCs and macroglia/Müller cells [1,10], while in retinal sections of the adult mouse retina, only Müller cells produced MMP2 [8,17]. After in vivo injury to the mouse optic nerve, MMP2 expression was occasionally associated with the growth cone of stimulated axons, yet reactive astrocytes at the site of the injury were identified as the major producers of MMP2 [18,20]. Furthermore, axonal injury to the spinal cord or optic nerve in adult rodents has been reported to result in a strong induction of gelatinase activity (e.g., MMP2 activity) in astrocytes within the scar tissue [18,20].

In line with its originally defined function as extracellular matrix (ECM)-remodeling enzyme, functional studies identified MMP2 as a key player in the resolution of the glial scar, which is needed for the regenerating axons to pass this physicochemical barrier. MMP2 has been shown to cleave inhibitory chondroitin sulphate proteoglycans (CSPGs) and other components of the glial scar [21]. After in vivo transplantation experiments of progenitor cells or peripheral nerve grafts in the injured CNS, a positive correlation was indeed found between MMP2 expression/activity by these cells, degradation of inhibitory components of the glial scar, and in vivo axonal regrowth [18,19,22,23]. Furthermore, olfactory ensheathing cell grafts, expressing very high levels of MMP2, reduced the CSPG levels in scar tissue in damaged rat spinal cords and, as such, promoted axonal regeneration [23]. Moreover, reducing the amount of CSPGs in the perineuronal nets by administering recombinant MMP2 to dissociated adult rat RGCs stimulated axonal regrowth [23]. Mechanistic insights into this glial scar-resolving role by MMP2 comes from ex vivo studies on retinal explants, showing that MMP2 acts at the growth cone and cleaves either the molecules interacting with β-integrin (e.g., cell adhesion molecule L1 and laminin) or β1-integrin directly, thereby stimulating axonal regeneration via a process closely resembling β1-integrin-mediated cell migration [10]. An in vitro study showed that, upon inhibition of MMP2 expression, immature astrocytes, which support neurite outgrowth and reduce scarring, were unable to cross an artificial inhibitory proteoglycan rim, and as such, axonal regrowth across the immature glial bridges was greatly reduced [24]. Moreover, *Mmp2^-/-^* mice were shown to exhibit increased glial scarring, causing an impaired structural and functional recovery after spinal cord injury [23]. All in all, these data suggest that MMP2 is a beneficial factor for axonal regeneration via ECM cleavage.

Apart from its function at the glial scar, MMP2 has also been reported to play a role during developmental axonal outgrowth and navigation. In vivo studies in *Xenopus* larvae demonstrated that inhibition of gelatinases—most likely MMP2—disrupt axonal navigation cues during retinofugal development when applied at low concentrations, whereas higher doses of the inhibitors also reduced axonal outgrowth [25,26]. Moreover, our research group provided primary proof that MMP2 is functionally involved in the growth of RGC axons towards the optic tectum in developing zebrafish [16,27]. These studies clearly revealed that gelatinases are required for proper development of the optic projection and may regulate RGC axon behavior at distinct decision points regardless of its role at the glial scar. These findings were further supported by in vitro experiments, reporting an increased neurite outgrowth in rat RGCs when co-cultured with glial cells that express MMP2 [23] and by ex vivo data from a mouse retinal explant model showing that MMP2 is necessary for robust neurite outgrowth [10]. The latter in-house generated data disclosed a co-involvement of MMP2 and MT1-MMP in axonal outgrowth and unveiled that MMP2, not MMP9, is required for optimal axon outgrowth of RGCs [10].

Altogether, MMP2 is considered a key player in axonal regeneration via either resolution of the suppressive extracellular environment or proteolysis of molecular cues that guide axon outgrowth. Here, we reveal a third mechanism via which MMP2 facilitates axonal regeneration, namely by orchestrating the expression of pro- and anti-inflammatory molecules by infiltrating innate immune cells. Starting from the novel finding that infiltrating myeloid cells abundantly express MMP2 in an in vivo inflammation-induced axonal regeneration mouse model, we reveal that MMP2 deficiency results in reduced long-distance axonal regeneration yet that this phenotype can be rescued by a WT bone marrow transplantation that restores MMP2 expression in the myeloid cells. Furthermore, MMP2 deficiency leads to qualitative rather than quantitative differences in this cell population, i.e., while the numbers of infiltrating immune cells remain unaltered, their inflammatory phenotype is different in MMP2 knock-out versus WT mice. In conclusion, we report a new facet of the pleiotropic role of MMP2 in axonal regeneration and provide evidence that MMP2 is an essential factor linking inflammation to axonal regeneration.

## 2. Materials and Methods

### 2.1. Experimental Animals

All animal experiments were approved by the Institutional Ethical Committee of KU Leuven and were conducted in strict accordance with European and Belgian legislation. Mice were housed in temperature-, humidity-, and light-controlled rooms with a 12 h light/dark cycle and had ad libitum access to food and water. All experiments were performed using 8–12-week-old C57BL/6 WT and MMP2-deficient (*Mmp2^-/-^*) mice [28]. In addition, bone marrow transplantation experiments were performed using 8–12-week-old Ly5.1 (CD45.1^+^) mice [29].

### 2.2. Surgical Procedures

#### 2.2.1. Heterologous Bone Marrow Transplantations

Heterologous bone marrow transplantations were performed as previously described [30]. Briefly, recipient mice (CD45.2^+^ *Mmp2^-/-^*, CD45.1^+^ WT, or CD45.2^+^ WT) were lethally irradiated the day before the transplantation with 900 Rad (9 Gy), divided over 2 equal doses of 450 Rad (4.5 Gy) separated by a time window of 4 h (RS-2000 Biological Irradation RAD source). Bone marrow cells of donor mice (CD45.2^+^ *Mmp2^-/-^* or CD45.1 WT) was collected from both the femur and tibia as previously reported [30]. Bone marrow cells were filtered and washed. A total of 1 × 10^6^ bone marrow cells, resuspend in 250 µL of Ham’s F12 medium (Gibco), were transplanted into each irradiated recipient by tail vein injection. CD45.2^+^ *Mmp2^-/-^* donor cells were transplanted into CD45.2^+^ *Mmp2^-/-^* or CD45.1^+^ WT recipients, whereas the CD45.1^+^ WT donor cells were injected into *Mmp2^-/-^* or CD45.2^+^ WT recipients. All of the recipient mice were allowed to recover for 4 weeks, before being subjected to follow-up treatments.

Percentage engraftment of the CD45.1^+^/CD45.2^+^ cells was determined at week 4 by analysis of the peripheral blood by flow cytometry (Appendix A). Briefly, blood was collected from the tail vein of each recipient mice. Thirty microliters of heparinized blood was stained with a six-color antibody panel (CD45.2-eFluor 450, CD45.1-FITC, CD11b-PE, 1/40, eBioscience). After staining, the blood samples were lysed and fixed using the 1-Step Fix/Lyse Solution (eBioscience) and acquired on the BD FACS CANTO II (BD Biosciences). FACS data were analyzed using FlowJo v10 software. Single live cells were first selected based on forward scatter and side scatter parameters. Then, the engraftment contribution was analyzed based on CD45.1^+^ and CD45.2^+^ expressions in total myeloid cells (CD11b^+^).

#### 2.2.2. Intraorbital Optic Nerve Crush Model

Optic nerve crush (ONC) was performed as previously described [3,31,32,33]. Briefly, an incision in the temporal side of the conjunctiva was made. Then, rotating the globe nasally, the posterior side of the eye was exposed, allowing visualization of the optic nerve. The exposed optic nerve was crushed approximately 1 mm from the optic nerve head with cross-action forceps for 5 s. Fundoscopy was performed to confirm blood supply to the eye, and animals with ischemia were excluded. Eyes from naive mice were used as controls.

#### 2.2.3. Intravitreal Injection

Pam3Cys (Sigma) combined with chlorophenylthio-cAMP (CPT-cAMP; cAMP analogue; Sigma) was injected intravitreally at a final concentration of 2.5 µg/mL [34] and 50 µM, respectively. Fluorescent conjugates of cholera toxin subunit B (CTB; Sigma) were injected intravitreally at a final concentration of 5 µg/mL. Intravitreal injections were performed as previously described [33,35]. Briefly, a syringe (Hamilton) equipped with a 34G needle (Hamilton) was used to inject either 2 µL of Pam3Cys and cAMP or 1 µL of CTB. The needle was inserted into the nasal part of the eye, at the limbus, under a 45° angle to avoid damage to the lens.

### 2.3. Immunohistochemistry on Retinal Whole Mounts

Mice were euthanized (pentobarbital, 60 mg/kg body weight, Nembutal, Vetoquinol) and transcardially perfused with saline and 4% phosphate buffered paraformaldehyde (4% PFA, pH 7.4). The eyes were dissected and post-fixed in 4% PFA at room temperature for 1 h. Thereafter, the retina was dissected and post-fixed in 4% PFA at room temperature for another hour. For immunostainings, whole mount retinas were incubated overnight with the primary antibody (goat anti-Brn3a; 1:500, sc-31984, Santa Cruz, diluted in 2% pre-immune serum and 2% triton X-100 (VWR) in PBS) at room temperature. Then, the samples were incubated with an Alexa Fluor^®^ conjugated secondary antibody for 2 h. Mosaic pictures of the entire retinal whole mounts were made using a laser scanning confocal microscope (Olympus FV 1000), and the maximum intensity projection images of z-stacks (step size of 3 µm) were analyzed. The total number of Brn3a^+^ RGCs was semi-automatically counted in complete retinal flat mounts using a validated Fiji algorithm and expressed as Brn3a^+^ cell number per mm^2^, all as previously described [32].

### 2.4. Immunohistochemistry on Retinal and Optic Nerve Cryosections

After perfusion, eye cups and optic nerves were post-fixed in 4% PFA for 1 h, embedded in TissueTek (Sakura), and 14 µm sagittal and longitudinal cryosections were made, respectively. Immunostainings for MMP2, GFAP, laminin, and CD45 were performed on consecutive retinal or optic nerve cryosections. Epitope retrieval was performed using citrate buffer (10 mM citric acid, 0.05% Tween 20, pH 6) using the pretreatment module (Thermofisher Scientific). Endogenous peroxidases were blocked by immersing in 0.3% H_2_O_2_ (in methanol) for MMP2 and CD45. After blocking with 20% pre-immune serum, the slides were incubated with the primary antibody (anti-MMP2 (1:200, sc-53630, Santa Cruz), anti-GFAP (1:1000, Z0334, Dako), anti-laminin (1:100, L9393, Sigma), and anti-CD45 (1:100, #553076, BD)) overnight at RT. Next, sections were either incubated for 2 h with an Alexa Fluor^®^ conjugated secondary antibody (GFAP) or with HRP-conjugated (laminin) or biotin-conjugated secondary antibodies (MMP2 and CD45) followed by tyramid signal amplification as described in the manufacturer’s manual (TSA Cy3-Tyr, Thermofisher Scientific). Cell nuclei were counterstained with 4′,6-diamidino-2-phenylindole (DAPI; 1 µg/mL in PBS, Dako). Of note, for all staining, negative controls were included by omitting the primary antibody and by replacing the primary antibody with IgG isotype control antibodies (Life Technologies) in the same dilution. Images of mid-sagittal retinal and optic nerve sections were made with an Olympus FV1000 confocal microscope. The size of the glial scar was analyzed as the area demarcated by GFAP^+^ astrocytes [36] or as the laminin^+^ area, normalized to the thickness of the optic nerve at the site of injury using Fiji.

### 2.5. Visualisation of Axonal Tracing

Z-stack images of the longitudinal optic nerve sections were taken with a confocal microscope (Olympus FV1000). Axon growth was quantified by manually counting the number of CTB^+^ axons every 150 µm (distance d) beyond the crush site, in 3–5 sections per optic nerve, using Fiji software. At each distance, the cross-sectional width of the nerve was measured along with counting the axon number. The total estimated number of axons in the optic nerve extending distance d from the ONC site was calculated using Equation (1), where r = radius of the nerve (150 µm) and t = thickness of the section (14 µm) [4].

(1)
Σad=πr2 . Average # axonsµm of nerve widtht


### 2.6. Flow Cytometry and Fluorescence-Activated Cell Sorting (FACS) of Myeloid Cells

The mice were euthanized and transcardially perfused with saline, and the retinas and optic nerves were transferred to ice-cold DMEM (Invitrogen). The vitreous was not removed from these retinas samples in order to include the infiltrating immune cells that were localized at the retina–vitreous interface. Cell dissociation was performed as previously described [37] using mechanical and enzymatic (10 U/mL collagenase I (Worthington), 400 U/mL collagenase IV (Worthington), and 30 U/mL DNase I (Worthington)) dissociations (3 × 10 min at 37 °C). Afterwards, the cells were filtered using a 70 µm nylon cell strainer, washed in MACS buffer (10 mL fetal calf serum (Thermofisher Scientific) and 2,5 mL ethylenediaminetetraacetic acid (EDTA, VWR) in 500 mL HBSS medium (Invitrogen)), and blocked with anti-mouse CD16/CD32 (clone 2.4 G2; 2µg/µL) (BD Biosciences). For flow cytometry, the cells were incubated with an eight-color antibody panel (F4/80-BV421, CD11c-BV510, Ly6G-FITC, Cx3cr1-PE, CD11b-PE/Cy7, Ly6C-APC, CD45-APC/Cy7, MHCII-PerCP/Cy5.5; 1:40; Biolegend). For FACS, cells were incubated with CD45-APC/Cy7, allowing for separation of CD45^+^ and CD45^-^ cells. Thereafter, samples were washed, centrifuged, and filtered before performing flow cytometry analysis using the BD FACS CANTO II (BD Biosciences) (for flow cytometry) or the SH800 (Sony) (for FACS). Flow cytometry data were analyzed with FlowJo v10 software. We adopted the gating strategy shown in the results: single live cells were selected based on cell size (forward scatter area, FSC-A) and complexity or granularity (side scatter, SSC-A). Leukocytes were selected based on CD45^+^ expression, and myeloid cells were gated on CD11b^+^ expression. Monocytes/monocyte-derived macrophages (mo/MF), neutrophils, macrophages, and microglia were distinguished based on CD45, CX3CR1, Ly6G, and Ly6C expression.

### 2.7. Quantitative Real-Time PCR (qRT-PCR)

The mice were euthanized, and the retinas and optic nerves were immediately snap-frozen. Total RNA was extracted using TRI reagent (Sigma) and the Nucleospin RNA II kit (Machery-Nagel). First-strand cDNA was synthesized using oligo-dT primers and Superscript III Reverse Transcriptase (Invitrogen), and quantitative real-time PCR was performed for *Tnf, Il1b, Ifng, Il10, Ym1, inos, Il6, Cntf, Mmp2,* and *Gap43* using SYBR green assays (Thermofisher Scientific). Sequences of these primers and the corresponding annealing temperatures are listed in Appendix A. *Top1* and *Ywhaz* were selected from a panel of 10 potential housekeeping genes using geNorm reference gene analysis [38]. All samples were run in duplicate with a CFX96 Touch Real-Time PCR detection system (BioRad), and a no template control and blank were added for each gene as negative controls. Thermal cycle parameters were 10 min at 95 °C and 40 cycles of 10 s at 95 °C and 60 s at the optimized annealing temperature (Appendix A). Normalized expression values were calculated using qBase software (Biogazelle). All values are represented as relative expression values normalized against the geometric mean of the reference genes.

### 2.8. Statistics

Statistical analyses were performed using GraphPad Prism 8 software (GraphPad Software). Normal distribution was evaluated using a Kolmogorov–Smirnov test and parallel equal variance between groups was tested. Outliers were identified and excluded based on a Grubb’s test (extreme studentized deviate method). All data are shown as mean ± standard error (SEM). Statistical tests are specified in the figure legends, together with the number of biologically independent samples (*n*) and/or technical repeats (N). Differences between two groups are considered statistically significant with a probability of α  <  0.05 for * *p* < 0.05, ** *p* < 0.01, *** *p* < 0.001, and **** *p* < 0.0001. Statistically significant differences between multiple groups are specified using different letters. Conditions with the same letter are not significantly different, while conditions with different letters are significantly different from each other.

## 3. Results

### 3.1. MMP2 Has a Beneficial Effect on Inflammation-Induced Axonal Regeneration

Since both ex vivo and in vivo data have shown that MMP2 plays a role in neurite outgrowth [10], we first investigated its contribution to RGC axonal regrowth in the injured optic nerve using an in vivo ONC model combined with inflammatory stimulation (IS) (this model is referred to as ONC + IS from here on). To study the effect of *Mmp2* deletion on axonal regeneration, the number of CTB^+^ axons was quantified in both WT and *Mmp2^-/-^* mice, at 14 and 42 days after ONC was combined with IS (dpi ONC + IS) (Figure 1A,B). At 14 dpi ONC + IS, we observed a decrease in the number of regenerating axons in optic nerves of *Mmp2^-/-^* mice compared to WT animals (Figure 1C).

After a longer post-injury period (42 dpi ONC + IS), this reduced capacity to regenerate had not been resolved (Figure 1D). At both time points, the reduction in the number of regenerating axons was most evident at locations close to the crush site, with the number of CTB^+^ axons in *Mmp2^-/-^* optic nerves 30–50% lower than in the WT optic nerves. This difference in intrinsic growth capacity was further explored by investigating the expression levels of growth associated protein 43 (Gap43) in naive retinas and at 4 dpi ONC + IS, in both WT and *Mmp2^-/-^* mice. We revealed an upregulation of GAP43 at 4 dpi ONC + IS compared to naive retinas in both genotypes; however, GAP43 upregulation was lower in the *Mmp2^-/-^* mice compared to WT mice. These findings corroborate previous studies suggesting that inflammatory treatment induces axonal growth initiation rather than elongation and, thus, imply that MMP2 is likely to affect initiation rather than elongation of axonal regeneration. Of note, we excluded that the decreased number of regenerating axons in *Mmp2^-/-^* mice was due to reduced RGC survival. No effect of genotype on the number of surviving Brn3a^+^ RGCs was observed in WT and *Mmp2^-/-^* animals at 14 dpi ONC + IS (Figure 2A–F), in line with that literature reporting that *Mmp2* deficiency does not Italic naffect neuronal survival after retinal ischemia-reperfusion injury [39].

Taken together, these data imply a beneficial effect of MMP2 on the number of RGCs that succeed in regenerating their axon.

### 3.2. The Pro-Regenerative Effect of MMP2 Is in Part Due to Resolution of the Glial Scar

To elucidate the reason for the reduced number of regenerating axons in *Mmp2^-/-^* mice, we first looked at the well-known function of MMP2 in glial scar remodeling. Optic nerves dissected at 14 dpi ONC + IS were stained for GFAP, an astrocytic marker, and for laminin, an ECM component of the glial scar and substrate of MMP2 (Figure 3A,B,D,E). Morphometric analysis revealed that, at 14 dpi ONC + IS, the glial scar, i.e., the area delineated by GFAP^+^ astrocytes or the laminin^+^ area at the crush site, tended to be larger in *Mmp2^-/-^* mice; albeit this difference was not statistically significant (Figure 3C,E).

As such, we can conclude that, in our model, the mechanism by which MMP2 supports inflammation-induced axonal regeneration can only be partially explained by resolution of the glial scar.

### 3.3. MMP2 Expression in the Infiltrating Inflammatory Cells Is Important for Axonal Regeneration

To further disentangle the observed pro-regenerative role of MMP2, we investigated the expression profile of MMP2 in the healthy and regenerating retina using immunohistochemical staining on retinal sections of WT mice at baseline and at 2, 7, and 14 dpi ONC + IS (Figure 4A–D).

MMP2 was found to be expressed in the Müller glia of healthy adult mouse (Figure 4A), as reported earlier [17], and this expression was found to be transiently upregulated at 7 dpi ONC + IS (Figure 4A–D). Notably, the inflammatory cells that infiltrate the vitreous cavity and retina upon IS and that were identified by positive immunostaining for the pan-leukocyte marker CD45 also showed prominent MMP2 expression (Figure 4E,F). These immunohistochemical data were confirmed by analyzing the mRNA expression levels of *Mmp2* in CD45^-^ (Müller glia, photoreceptors, RGCs, and other neurons) and CD45^+^ (infiltrating inflammatory cells and microglia) cells sorted from WT naive retinas and retinas at 4 dpi ONC + IS. Indeed, both cell populations express MMP2, and a clear upregulation is seen in CD45^-^ cells—probably in the Müller glia—in ONC + IS versus naive samples (Figure 4G). Of note, upregulation of MMP2 expression in CD45+ cells in ONC + IS versus naive samples could not be studied, since (almost) no infiltrating CD45+ cells are present in a naive retina. These expression and localization data thus link MMP2 to both Müller glia and infiltrating inflammatory cells after optic nerve injury with inflammatory treatment.

To elucidate the relative importance of the infiltrating inflammatory cells producing MMP2, heterologous bone marrow transplantations were performed. Bone marrow cells of WT (CD45.1) mice were transplanted in irradiated *Mmp2^-/-^* animals and vice versa (Table 1).

A replacement of the peripheral blood by the heterologous bone marrow-derived cells by more than 99% was shown by flow cytometry after 4 weeks post-transplantation (Appendix A). The bone marrow-engrafted mice were subjected to ONC + IS, and the number of CTB^+^ axons was quantified at 14 dpi ONC + IS. Transplanting WT bone marrow in *Mmp2^-/-^* mice rescued the regeneration defect that we previously observed in the *Mmp2^-/-^* mice. In other words, transplanting WT bone marrow in *Mmp2^-/-^* mice resulted in a number of regenerating axons equal to WT mice and WT mice transplanted with WT bone marrow. Vice versa, *Mmp2^-/-^* bone marrow transplants in WT mice led to a number of regrowing axons that are similar to those in *Mmp2^-/-^* mice and *Mmp2^-/-^* mice with *Mmp2^-/-^* bone marrow grafts (Figure 5). All in all, these results suggest that MMP2 produced by immune cells is needed to stimulate axonal outgrowth by the RGCs.

### 3.4. MMP2 Does Not Alter the Proliferation or Influx of Inflammatory Cells after Optic Nerve Crush Combined with Inflammatory Stimulation

As the bone marrow transplantation experiments show that MMP2 produced by the immune cells is important for inflammation-induced axonal regeneration, we studied whether an altered influx or proliferation of infiltrating and resident inflammatory cells could explain the reduced number of outgrowing axons in *Mmp2^-/-^* mice. Therefore, we performed a flow cytometry analysis of the myeloid cells infiltrating the retina and optic nerve of WT and *Mmp2^-/-^* animals at different time points after injury (naive and 2, 4, 6, and 8 dpi ONC + IS) (Figure 6A).

As the uninjured retina only contains a limited number of blood-borne inflammatory cells, the number of cells in the microglia/macrophage gate are mostly microglia in the naive samples. A very low but equal number of neutrophils, monocytes, and macrophages and a similar number of microglia were detected in the retinas as well in both genotypes. Upon ONC + IS, we observed a significant increase in neutrophils and inflammatory monocytes in the retina and optic nerve at 2 dpi ONC + IS, which again reduced at 4 dpi ONC + IS (Figure 6B,C,E,F). The microglia/macrophage cell population increased from 4 dpi ONC + IS in the retina and from 6 dpi ONC + IS in the optic nerve and remained upregulated until at least 8 dpi ONC + IS (Figure 6D–G). None of the myeloid cell populations differed in *Mmp2^-/-^* versus WT mice (Figure 6B–G). As such, the present data indicate that MMP2 deficiency does not affect the number of different innate immune cells in the retina or optic nerve upon ONC + IS.

### 3.5. MMP2 Does Alter the Expression Profile after Optic Nerve Crush Combined with Inflammatory Stimulation

Finally, we investigated whether MMP2 deficiency alters the phenotype rather than the absolute numbers of the innate immune cells. The mRNA expression levels of a panel of pro- and anti-inflammatory molecules (respectively *Tnf, Il1**β, Ifnγ, and Il10, Ym1,* and *Inos*) was analyzed in naive and regenerating retinas and optic nerves of both WT and *Mmp2^-/-^* mice. Retinal expression levels were upregulated for all molecules at 4 dpi ONC + IS in both genotypes (Figure 7).

Remarkably, the pro-inflammatory cytokines T*nf* and *Il1**β* (and to a lesser extent *Ifnγ*) showed a reduced upregulation in the retina of *Mmp2^-/-^* mice (Figure 7A,B), while the anti-inflammatory molecule *Ym1* (Figure 7D) showed a higher upregulation in the *Mmp2^-/-^* retinas compared to WT samples. On the other hand, in the optic nerve, all molecules measured showed similar expression levels at 4 dpi ONC + IS in WT and *Mmp2^-/-^* retinas (Figure 7G–L). Overall, the retina of *Mmp2^-/-^* mice subjected to ONC + IS seems to be characterized by a phenotypic switch, from a pro- to an anti-inflammatory expression profile, in comparison to WT mice. This differential expression of pro- and anti-inflammatory molecules, which is evident in the retina yet not in the optic nerve, suggests a role for MMP2 in orchestrating inflammatory cues in the retina that instruct inflammation-induced axonal regeneration in the optic nerve.

Importantly, besides microglia and immune cells, macroglia are also involved in the inflammatory response. To determine the contribution of the macroglia during inflammation-induced axonal regeneration, we looked at the expression profile of IL6 and CNTF, two pro-regenerative factors that are produced by the Müller glia [40,41], in the retina of WT and *Mmp2^-/-^* mice. The expression levels of IL6 and CNTF were upregulated at 4 dpi ONC + IS compared to naive retinas in both genotypes (Figure 8).

However, IL6 and, to a lesser extent, CNTF were upregulated more in the WT mice compared to the *Mmp2^-/-^* mice (Figure 8). This suggests that MMP2 influences the production of these Müller glia-derived pro-regenerative molecules and thus links to both macroglia and innate immune cell responses in the retina.

## 4. Discussion

In this study, we sought to explore the role of MMP2 during inflammation-induced axonal regeneration in the mouse optic nerve. Using a well-established regenerating optic nerve injury model in which ONC is combined with the induction of an acute inflammatory response in the retina (ONC + IS), we observed a diminished RGC axonal regeneration in *Mmp2^-/-^* mice compared to WT animals. These findings confirm previous studies on the contribution of MMP2 to a successful regeneration of RGC axons upon crush injury [10,23,25,26]. What is new in this study is that we identified a novel mechanism via which MMP2 promotes axonal regeneration. Indeed, while we confirm that MMP2 remodels the glial scar and therefore facilitates axonal regrowth, we also point out that it is an essential factor linking inflammation to successful regeneration. MMP2 levels were found to be upregulated in infiltrating myeloid cells, besides Müller glia, in our regenerative ONC model (ONC + IS). This expression appears to be required for axonal regeneration in this model. Based on flow cytometry analyses and expression profiling of the innate immune cell populations, we hypothesize that MMP2 does not affect the number of myeloid cells after ONC + IS but that MMP2 produced by infiltrating leukocytes, either directly or indirectly, upregulates pro-inflammatory and pro-regenerative molecules while suppressing anti-inflammatory factors in the retina. Altogether, the reduced axonal regeneration seen in *Mmp2^-/-^* mice in this study is likely the combined result of the roles of MMP2 in glial scarring and in the regulation of the inflammatory expression profile of innate immune cells.

As outlined in the Introduction, MMP2 is mainly known for its extrinsic role in the proteolysis of ECM components and guidance molecules at the glial scar and along the axon trajectory, thereby clearing the path for axons to regrow, providing navigation cues, and as such stimulating axonal regeneration [18,19,21,22,23,24]. In this study, however, the difference in size of the glial scar in WT and *Mmp2^-/-^* mice was too small to explain the difference in axonal regeneration that was observed between the two genotypes. This suggests that there is an additional complementary underlying mechanism via which MMP2 stimulates axonal regeneration. This novel function of MMP2 seems to be based on its expression by infiltrating myeloid cells. Indeed, here, we revealed an additional cell type in the mammalian retina/eye, besides the previously described expression by macroglia/Müller cells and RGCs and their axons [1,8,17,42], that abundantly expresses MMP2 upon inflammation-induced axonal regeneration. This expression pattern, further supported by the notion that regenerating zebrafish RGC axons upregulate their MMP2 expression despite growing in a permissive environment devoid of glial scarring [27] and that this MMP2 upregulation is essential for RGC axonal regrowth upon ONC [11,43], suggests that MMP2 is involved in pro-regenerative processes other than breaking down the glial barrier.

We thus investigated whether MMP2 may be a modulator of the inflammatory response that, by now, is well-known to instruct axonal regeneration. Heterologous bone marrow transplantations confirmed that MMP2 expression by the innate immune cells is essential for successful axonal regeneration. Furthermore, we show that MMP2 plays a central role in the orchestration of inflammatory response in the inflammation-induced axonal regeneration model used in this study. First, we assessed a role for MMP2 in facilitating the influx or proliferation of innate immune cells. Other MMPs, e.g., MMP3 and MMP9, have been suggested to modulate blood retinal barrier integrity after injury/damage and to therefore regulate the influx of inflammatory cells [3,44,45,46]. However, our flow cytometry data exclude that a difference in the number of cells in various myeloid cell populations, in the retina nor the optic nerve, between WT and *Mmp2^-/-^* mice underlies the abrogated axonal regeneration in *Mmp2^-/-^* mice. Thus, in a second series of experiments, we investigated qualitative rather than quantitative differences in the immune cells of WT and *Mmp2^-/-^* mice subjected to ONC + IS. Since the beneficial effect of MMP2 on axonal regeneration cannot be explained by a differential number of infiltrating myeloid cells in the *Mmp2^-/-^* mice compared to WT animals, we hypothesized that a differential expression pattern of inflammatory and regenerative molecules by either the infiltrating/resident myeloid cells and/or the macroglia might lie at the basis of the observed phenotype. Strikingly, within the optic nerve, the expression of both pro- and anti- inflammatory molecules did not change after ONC + IS. These data provide additional evidence that, in our model, MMP2 does not facilitate axonal regeneration by altering inflammatory/glial reactivation at the injury site or by degrading ECM components of the glial scar. In contrast, in the retina, we revealed a reduced pro-inflammatory (*Il1**β* and *Tnf*) and an increased anti-inflammatory (*Ym1*) expression profile in *Mmp2^-/-^* animals compared to WT mice when subjected to ONC + IS. These data indicate a pro-regenerative effect of (a transient upregulated expression of) IL1β and TNF and contrast previous reports in mammals, in which *Il1**β*-deficient mice were characterized by a smaller lesion size, less astrogliosis, and a higher number of regenerating axons after spinal cord compression injury, suggesting IL1β has a negative effect on axon outgrowth, at least at the site of injury [47,48]. The effect of TNF on axonal regeneration is not clear in mammals and may be either promoting [49,50] or inhibiting [51]. Interestingly, however, using a spinal cord injury paradigm in spontaneously regenerating zebrafish, Tsarouchas et al (2019) reported that axonal regeneration is dependent on an initial transient upregulation of Il1β, followed by a later increase in Tnf [52]. Successful axonal regeneration thus apparently needs a biphasic inflammatory response with an initial pro-inflammatory phase in which IL1β and TNF play crucial roles [48,52,53]. Although we cannot provide conclusive evidence at this time, the imbalance in the (transient) upregulations of IL1β and TNF in *Mmp2^-/-^* mice might thus underlie the observed reduction in the number of outgrowing axons. We realize that TNF and IL1b are certainly not the only molecules contributing to the decrease in axonal regrowth in this *Mmp2^-/-^* mouse model, in which compensatory mechanisms due to Mmp2 knockout may further complicate these findings. Identifying the full spectrum of differentially expressed genes in WT and Mmp2^-/-^ mice contributing to inflammation-induced axonal regeneration could shed light on molecular pathway changes in neurons, glial cells, and myeloid cells or even in their cross talk. We also looked at the expression profile of two well-known pro-regenerative molecules that are produced by the macroglia, namely IL6 and CNTF. Previous studies have shown that intravitreal injection of IL6 enabled axon regeneration beyond the ONC site [54] and that genetic ablation of IL6 in mice significantly reduced inflammation-induced axonal regeneration in the optic nerve [54]. Similarly, the neuroprotective and axon growth-promoting effects of lens injury were significantly reduced in CNTF-deficient mice compared with WT animals [55]. Here, we show that the absence of MMP2 expression goes hand in hand with a reduced upregulation of IL6 and CNTF and that this may underlie the reduced regenerative capacity of *Mmp2^-/-^* mice.

Our data suggest that MMP2 contributes to inflammation-induced axonal regeneration by activation and/or suppression of certain molecules. Since in our model, the inflammatory stimulation is situated in the retina and not in the optic nerve, it seems logical that the pro-regenerative effect of MMP2 upon ONC + IS manifests in the retina and not in the optic nerve at the glial scar. One forthcoming hypothesis is that, in the retina, MMP2 produced by the inflammatory myeloid cells may, directly or indirectly, activate the Müller glia. Together with the innate immune cells, these activated Müller glia produce other molecules (e.g., CNTF and IL6) as well as increased levels of MMP2, which could in turn activate certain intrinsic growth-inducing pathways in the RGCs. Clearly, more research is needed to pinpoint the cellular and molecular players contributing to inflammation-induced optic nerve regeneration in the adult mouse visual system and how these might be affected by MMP2. Future experiments combining cell sorting of the specific myeloid and macroglia cells with qPCR or bulk RNA sequencing at multiple time points can help resolve the exact cells and underlying pathways, and their respective timing, via which MMP2 connects inflammation to successful axonal regeneration.

## Figures and Tables

**Figure 1 cells-10-01672-f001:**
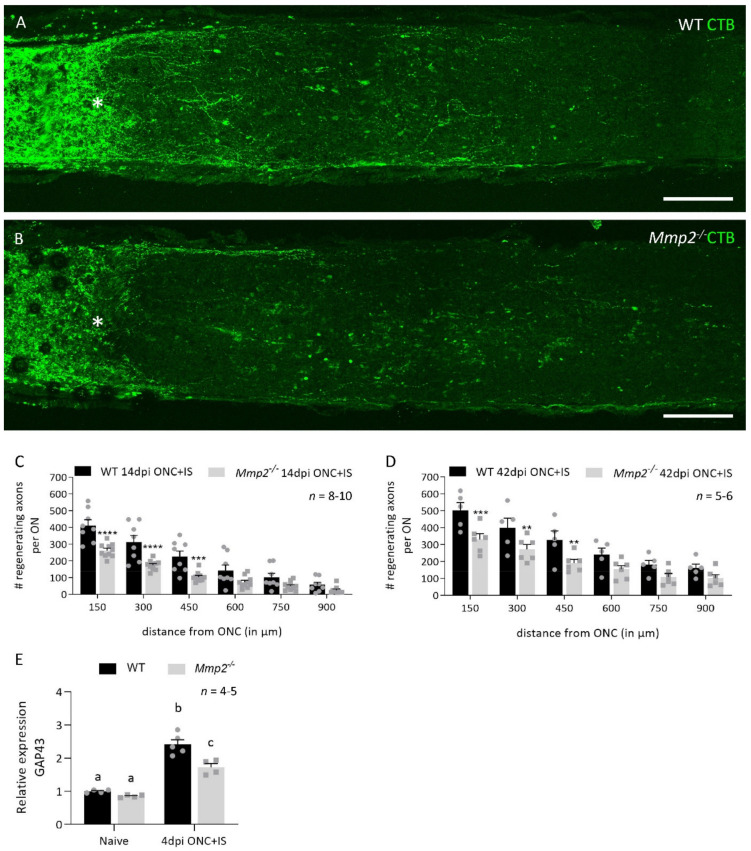
*Mmp2^-/-^* mice have a reduced number of regenerating axons compared to WT mice. (**A**,**B**) The number of regenerating axons visualized using CTB tracing in WT (**A**) and *Mmp2^-/-^* (**B**) mice at 14 dpi ONC+IS. The ONC site is indicated with an asterisk. Scale bar: 100 µm. (**C**) Graphic of the number of CTB^+^ axons per optic nerve in function of the distance from the ONC site comparing the number of CTB^+^ axons in WT (black) and *Mmp2^-/-^* (grey) mice at 14 dpi ONC+IS. The number of CTB^+^ axons was counted every 150 µm from the ONC site till 900 µm. (**D**) Graphic of the number of CTB^+^ axons per optic nerve in function of the distance from the ONC site comparing WT (black) and *Mmp2^-/-^* (grey) mice at 42 dpi ONC+IS. (**E**) qRT-PCR data showed the expression of GAP43 at 4 dpi ONC+IS in WT and *Mmp2^-/-^* mice. Data are relative expression values compared to the expression value in naive WT retinas (fold change). Data are shown as mean ± SEM. Repeated measures two-way ANOVA for axonal regeneration analysis followed by Sidak’s multiple comparisons test, * *p* < 0.05, ** *p* < 0.01, *** *p* < 0.001, **** *p* < 0.0001 for comparisons between genotypes. Repeated measures two-way ANOVA for qRT-PCR analysis followed by Tukey’s multiple comparisons test. Statistical significance between different time points is indicated using different letters.

**Figure 2 cells-10-01672-f002:**
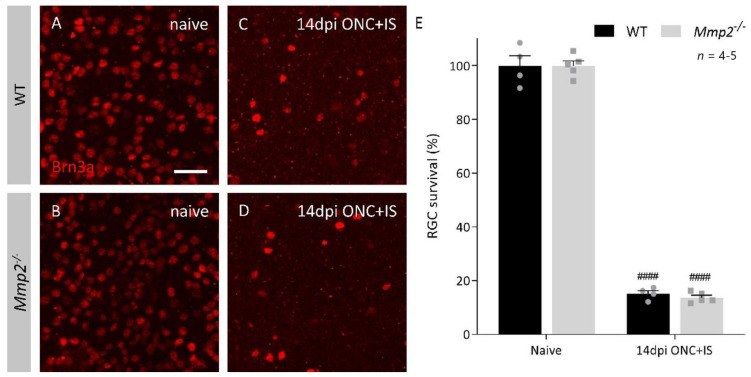
RGC survival in WT and *Mmp2^-/-^* mice at 14 dpi ONC+IS in whole mount retinas. (**A**–**D**) Representative images of Brn3a immunostaining on retinal wholemounts in WT and *Mmp2^-/-^* mice disclosed around 80% loss of Brn3a^+^ RGCs at 14 dpi ONC+IS as compared to naive mice. Scale bar: 50 µm. (**E**) Quantitative analysis on wholemounts revealed no significant difference between the number of Brn3a^+^ RGCs surviving at 14 dpi ONC+IS in the *Mmp2^-/-^* mice (grey) compared to WT mice (black). Data are shown as mean ± SEM. Repeated measures two-way ANOVA for RGC survival followed by a Tukey’s multiple comparison test between WT and *Mmp2^-/-^* mice and comparing naïve and 14 dpi ONC+IS per genotype, ^####^ *p* < 0.0001 for comparisons with baseline.

**Figure 3 cells-10-01672-f003:**
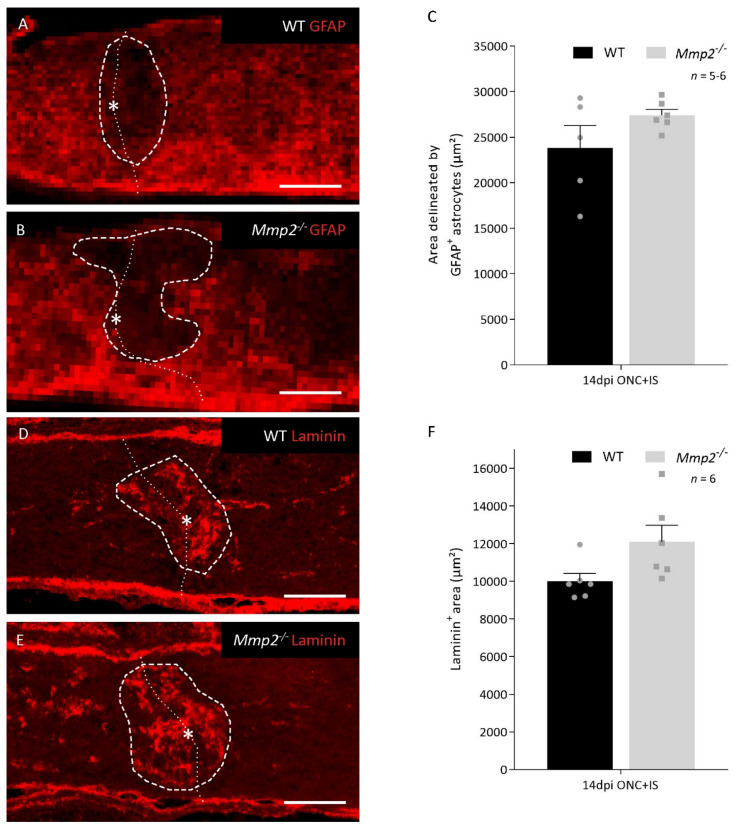
Glial scar formation in WT and *Mmp2^-/-^* mice at the optic nerve injury at 14 dpi ONC+IS. (**A**,**B**) Images showing a GFAP immunostaining at the ONC site for WT (**A**) and *Mmp2^-/-^* (**B**) mice at 14 dpi ONC+IS. The ONC site is indicated with an asterisk and a dotted line and the measured area is indicated with a dashed line. Scale bar: 100 µm. (**C**) Graphic of the quantification of the glial scar size measured as the area delineated by GFAP^+^ astrocytes at the ONC site in WT (black) and *Mmp2^-/-^* (grey) mice at 14 dpi ONC+IS. The area was normalized to the width of the optic nerve. (**D**,**E**) Images showing a Laminin immunostaining at the ONC site for WT (**D**) and *Mmp2^-/-^* (**E**) mice at 14 dpi ONC+IS. The ONC site is indicated with an asterisk and a dotted line and the measured area is indicated with a dashed line. Scale bar: 100 µm. (**F**) Graphic of the quantification of the glial scar size measured as the Laminin^+^ area at the ONC site in WT (black) and *Mmp2^-/-^* (grey) mice at 14 dpi ONC+IS. The area was normalized to the width of the optic nerve. Data are shown as mean ± SEM. Statistical significance was determined by Student’s *t*-test.

**Figure 4 cells-10-01672-f004:**
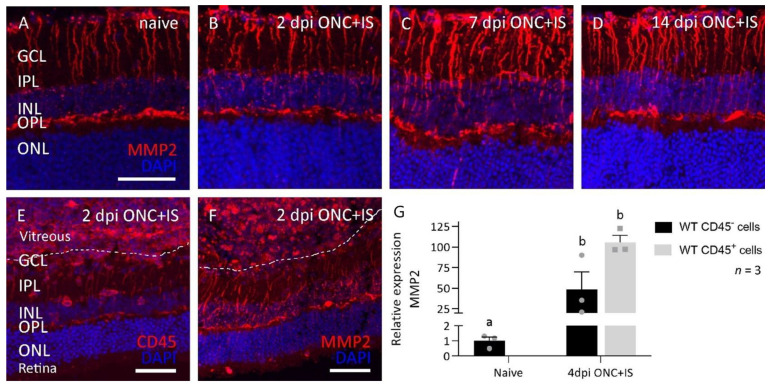
Spatiotemporal expression pattern of MMP2 in the eye of WT mice subjected to the ONC+IS model. (**A**–**D**) Retinas are dissected at 2, 7 and 14 dpi ONC+IS together with naive retinas. Representative images of MMP2 expression in the Müller glia show an upregulation of MMP2 at 7dpi ONC+IS which goes back to baseline at 14dpi ONC+IS. (**E**,**F**) Inflammatory stimulation cause an influx of CD45^+^ cells (**E**) into the vitreous body and retina, which prominently express MMP2 (**F**). Scale bar: 50 μm, GCL: ganglion cell layer, IPL: inner plexiform layer, INL: inner nuclear layer, OPL: outer plexiform layer, ONL: outer nuclear layer. (**G**) qRT-PCR data showed the expression of MMP2 at 4 dpi ONC+IS in both sorted CD45^-^ and CD45^+^ cells. Data are relative expression values compared to the expression value in naive WT CD45^-^ cells (fold change) and presented as mean ± SEM. Repeated measures two-way ANOVA for qRT-PCR analysis followed by Tukey’s multiple comparisons test. Statistical significance between different time points is indicated using different letters.

**Figure 5 cells-10-01672-f005:**
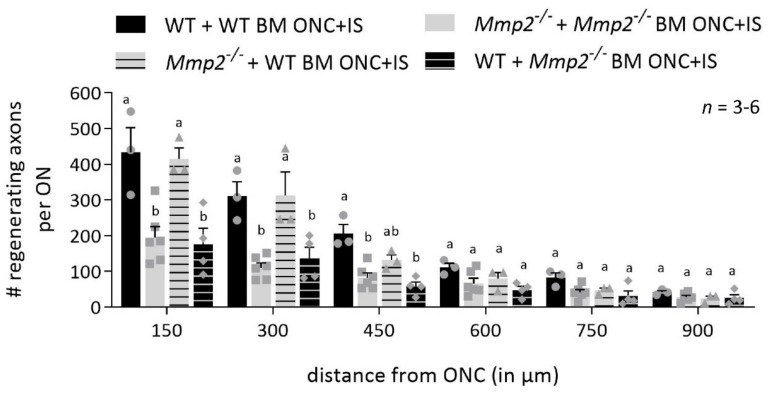
*Mmp2^-/-^* mice transplanted with WT bone marrow show a rescue effect. Graphic of the number of CTB^+^ axons per optic nerve in function of the distance from the ONC site, comparing the number of CTB^+^ axons in WT mice transplanted with WT BM (black), *Mmp2^-/-^* mice with *Mmp2^-/-^* BM (grey), *Mmp2^-/-^* mice with WT BM (grey with black stripes) and WT mice with *Mmp2^-/-^* BM (black with grey stripes) at 14 dpi ONC+IS. The number of CTB^+^ axons was counted every 150 µm from the ONC site till 900 µm. At closer distances from the ONC site, there was higher number of CTB^+^ axons in the WT + WT BM mice compared to the *Mmp2^-/-^*+ *Mmp2^-/-^* BM mice, confirming data in Figure 1. However, similar numbers of CTB^+^ axons were observed in WT + WT BM mice and *Mmp2^-/-^* + WT BM mice, indicating a rescue effect of WT bone marrow transplanted in *Mmp2^-/-^* mice. Strikingly, the WT+ *Mmp2^-/-^* BM mice show similar numbers of CTB^+^ axons as in *Mmp2^-/-^*+ *Mmp2^-/-^* BM animals, suggesting a clear contribution of MMP-2 expressing leukocytes in initiating axonal regrowth. Data are shown as mean ± SEM. Repeated measures two-way ANOVA for axonal regeneration analysis followed by Sidak’s multiple comparisons test, statistical significance between different conditions is indicated using different letters.

**Figure 6 cells-10-01672-f006:**
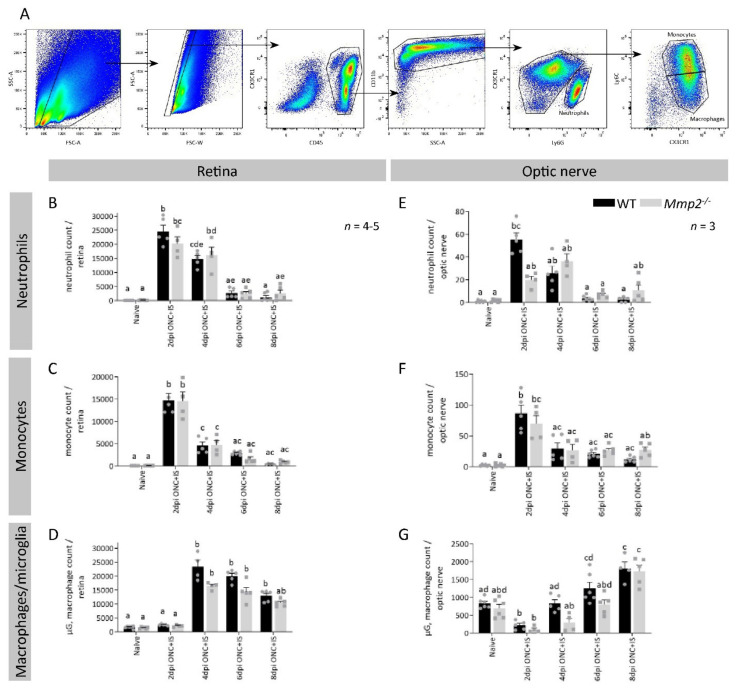
The effect of MMP2 deficiency on the resident and infiltrating myeloid cells in the retina and optic nerve. (**A**) Gating strategy applied to all retinal and optic nerve samples for both genotypes in naive mice and at 2, 4, 6 and 8 dpi ONC + IS. The single cell population was identified based on cell size and complexity/granularity (respectively FSC and SSC). First, all leukocytes were gated based on CD45 expression and then the myeloid cells were selected based on CD11b expression. The different myeloid cell populations of interest were identified as neutrophils (referred to as Ly6G^+^, CX3CR1^-^), monocytes (referred to as Ly6G^-^, CX3CR1^+^, Ly6C^+^) and microglia/macrophages (referred to as Ly6G^-^, CX3CR1^+^, Ly6C^-^). (**B**–**G**) Compiled flow cytometry data show similar trends for all cell populations at all time points between Mmp2-/- and WT mice in both the retina (**B**–**D**) and optic nerve (**E**–**G**). Data are shown as mean ± SEM. Repeated measures two-way ANOVA followed by Tukey’s multiple comparisons test, statistical significance between different time points and conditions is indicated using different letters.

**Figure 7 cells-10-01672-f007:**
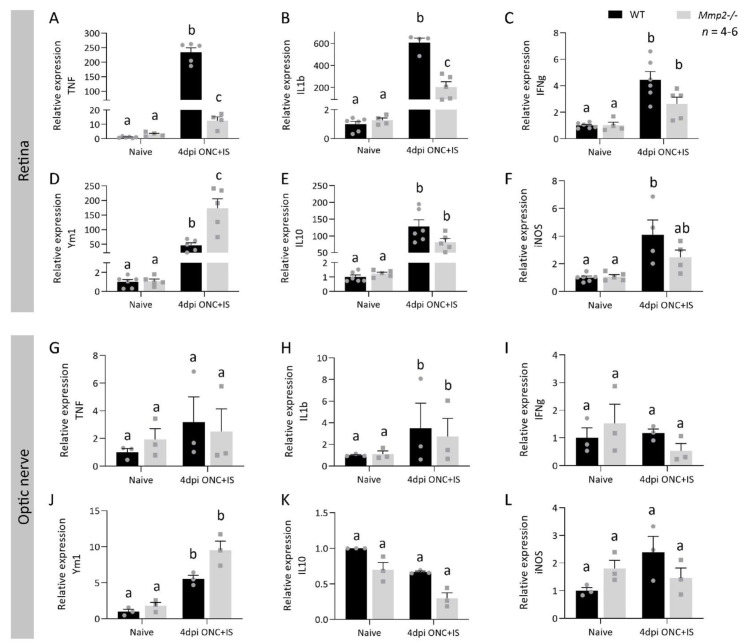
MMP2 deficiency affects the expression profile of pro- and anti-inflammatory cytokines in the retina and optic nerve after ONC injury combined with an inflammatory stimulation. qRT-PCR data showed the expression of pro- (**A**–**C**,**G**–**I**) and anti-inflammatory cytokines (**D**–**F**, **J**–**L**) at 4 dpi ONC+IS in retinas (**A**–**F**) and optic nerves (**G**–**L**) of both WT mice and *Mmp2^-/-^* mice. Data are relative expression values compared to the expression value in naive WT retinas or optic nerves (fold change) and presented as mean ± SEM. Repeated measures two-way ANOVA for qRT-PCR analysis followed by Tukey’s multiple comparisons test. Statistical significance between different time points is indicated using different letters.

**Figure 8 cells-10-01672-f008:**
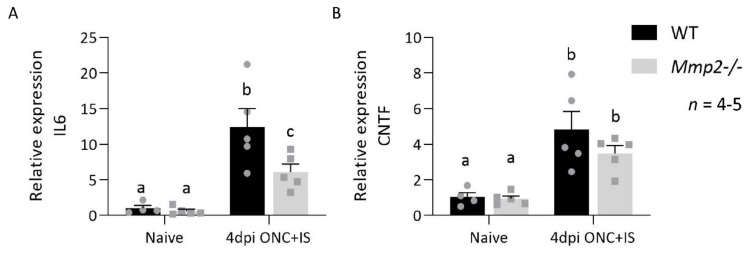
MMP2 deficiency affects the expression profile of IL6 and CNTF in the retina after ONC injury combined with an inflammatory stimulation. qRT-PCR analysis of the expression of IL6 (**A**) and CNTF (**B**) at 4 dpi ONC+IS in retinas of WT mice and *Mmp2^-/-^* mice. Data are relative expression values compared to naive WT retinas (fold change) and presented as mean ± SEM. Repeated measures two-way ANOVA for qRT-PCR analysis followed by Tukey’s multiple comparisons test. Statistical significance between different time points is indicated using different letters.

**Table 1 cells-10-01672-t001:** Schematic overview of the different conditions for bone marrow transplantations and the resulting cellular genotypes. BM: bone marrow, WT: wild-type.

Condition	Explanation	Müller Glial	Infiltrating Inflammatory Cells
WT + WT BM	CD45.1+ bone marrow in CD45.2+ mice	WT	WT
Mmp2-/- + Mmp2-/- BM	CD45.2+ Mmp2-/- bone marrow in CD45.2+ Mmp2-/- mice	Mmp2-/-	Mmp2-/-
WT + Mmp2-/- BM	CD45.2+ Mmp2-/- bone marrow in CD45.1+ mice	WT	Mmp2-/-
Mmp2-/- + WT BM	CD45.1+ bone marrow in CD45.2+ Mmp2-/- mice	Mmp2-/-	WT

## Data Availability

The data sets generated and analyzed during the current study are available from the corresponding author upon reasonable request.

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
