# Peer review of "MMP2 Modulates Inflammatory Response during Axonal Regeneration in the Murine Visual System"

_cells, 2021, doi:10.3390/cells10071672_

Round 1

Reviewer 1 Report

In this manuscript entitled “MMP2 modulates the inflammatory response during axonal regeneration in the murine visual system”, Andries and colleagues studied the role of MMP2 in inflammation-induced axonal regeneration in the mouse optic nerve lesion paradigm. Indeed, in different growth conditions, MMP2 expression is relatively high in neurons. In the retina, the authors have previously contributed to show that MMP2 is endogenously expressed in retinal ganglion cells and in Mueller glia (Macroglia). Strikingly, they report here that MMP2 gene ablation significantly reduced optic nerve axon growth following inflammation-induced activation of ganglion cell growth. This effect was only weakly associated with glial scar change at the injury site, in the optic nerve, as shown with GFAP and laminin staining. However, by transplanting bone marrow from WT mice to MMP2 KO animals, the authors showed a rescue effect, thus suggesting the involvment of MMP2 in myeloid cells. Moreover, their data suggest a significant downregulation of IL1b and TNF, two inflammatory molecules induced in the retina. Together, these data suggest that MMP2 expression in myeloid cells is required to activate axonal regeneration in the injured adult optic nerve. This might depend on some inflammatory cytokines.

This study reveals a positive role of MMP2 in optic nerve axon regrowth after injury. The methods used allow to determine with a very high degree of accuracy the level of axonal growth and the changes caused by MMP2 removal. It is also not trivial to observe a significant contribution of MMP2 expression in immune cells, to retinal neuron regrowth. In this regard, the work of Andries et al sheds a new light on the mechanisms of axonal regeneration/neuronal growth after traumatic lesion.

My additional suggestions and comments to improve the manuscript are the following:

  • I wondered if the authors had the opportunity to examine MMP2 expression changes in myeloid cells after inflammation stimulation (qRT-PCR or Western blot). I think that this may be interesting to show, if the authors have samples in reserve for this (e.g. after FAC sorting)
  • Caution must be exerted with the observation of TNF and IL1b downregulation in MMP2 KO retinae as the expression of many other genes may contribute to axonal growth decrease in conventional KO, and given that MMP2 expression can be found in neurons, glia and immune cells. Moreover, I would like to suggest the examination of neuronal growth gene expression, such as GAP43, in order to address intrinsic growth mechanism changes. If possible.
  • Statistics with letters, a, b, c are really not clear in many figures of the manuscript. Please, change this thoroughly.
  • It is important to define “IS”. Please, limit the use of acronyms to facilitate the reading. I can only guess that it means inflammation stimulation.
  • I find that the selected astrocyte-free area in Fig 3B could be more carefully defined as it seems to me larger than what the selected region suggests.
  • The authors wrote that optic nerve lesion was carried out at 1 mm from the optic nerve head. It is relatively distal. Please, check.

Reviewer 2 Report

Quite nice data, providing new insights on the MMP2 function. I have a few comments:

1. Detailed description for quantifying the cell numbers such as RGC survival and astrocytes.

2. Negative control for immunostaining

3. Ideally measuring cytokines using ELISA.

Reviewer 3 Report

In this study, researchers explored the role of MMP2 using ONC model and observed a diminished RGC axonal regeneration in Mmp2 knock out mice.

below are the concerns need to be addressed:

  1. The introduction part is too long and must be shortened. A concise review of MMP2 research will be enough to give readers the study background.  
  2. what does IS represent?
  3. Have you co-stained for both CD45 and MMP2 on ONC+IS tissue? Do you see them co-localized on the tissue? from the image in Figure 4, it is not quite clear that the cells were positive for both CD45 and MMP2. Also, It would be nice to have quantification data in Figure 4 to show the MMP2 is increased.
  4. what do those a,b,ab mean in figure 5,6,7,8 captions?

Round 2

Reviewer 2 Report

The authors have addressed my comments. I have no further concern.

Reviewer 3 Report

concerns are well addressed.